# GRM-Omni: Generative Omni-modality Reward Modeling via Meta Reward Learning

## Abstract

Scaling from sparse and underspecified rewards is a core challenge for the continuous improvement of foundation models in decision-making. Reward modeling and policy optimization have traditionally been decoupled, which often results in policies overfitting static reward models and thus limits the scalability and generalization. In this work, we propose a **meta-reward learning** algorithm that couples discriminative and generative reward models with policy models, producing *scalable intrinsic rewards* that bridge the gap between sparse environmental rewards and the dynamics of policy learning. The goal of meta-reward learning is to train a reward model capable of generalizing effectively across diverse scenarios under limited supervision, such as handling unseen modalities or tasks. In particular, our dual-reward design can attribute each scalar reward to multiple underlying language criteria and iteratively refine their priority, thereby enabling continuous improvement of both policy and reward. We implement GRM-Omni, an omni-modal reward model that not only achieves strong results on multiple multi-modal preference reward benchmarks but also facilitates more effective policy decisions.

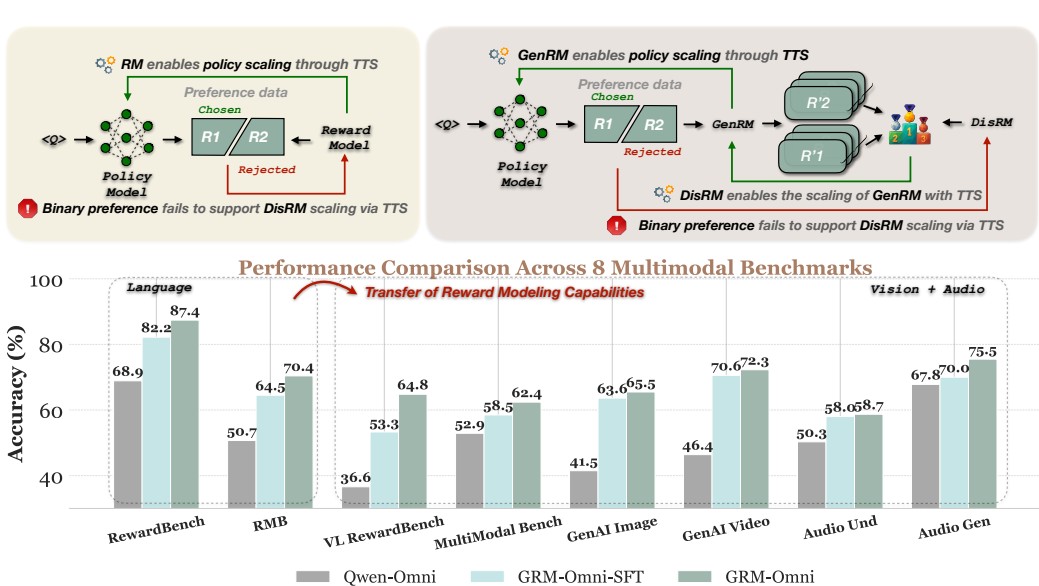

Figure 1: Traditional approaches to policy optimization and reward modeling often face limitations in scalability and generalization, primarily due to the high cost of collecting large-scale preference data (red line). To overcome these challenges, we introduce a GenRMs that mitigate the learning rate gap between policy and reward models (green line). A key component of our framework is a novel meta-reward formulation, designed to alleviate compounding bias in reward modeling. Based on this, we implement GRM-Omni, which substantially enhances reward modeling across modalities. In particular, we show that our trained model solely on language-based preference data can generalize effectively to multimodal secnrios by leveraging criteria-based TTS, highlighting the effectiveness of our meta-reward learning algorithm.

# 1 INTRODUCTION

Under the Reinforcement Learning (RL) paradigm (Sutton et al.; OpenAI et al., 2024; Guo et al., 2025a), Large Language Models (LLMs) (Grattafiori et al., 2024; Yang et al., 2025; Team et al., 2025) can leverage reward models to transform environmental or human feedback into gradient signals for updating model parameters. Reward Models (RMs) provide a proxy objective for policy learning and effectively mitigate the *sparsity* and *delay* of environmental rewards. As one of the most widely used models, the Bradley–Terry (BT) (Bradley & Terry, 1952) model learns a scalar utility score from pairwise preferences; however, it still faces two main challenges:

➡**1:** *Due to Goodhart's law, scalar rewards often fail to provide precise feedback. As a result, policy models are prone to overfitting the proxy signal and vulnerable to reward hacking (Pan et al., 2022; Skalse et al., 2025; Langosco et al., 2023; Everitt et al., 2021).*

➡**2:** *Their reliance on human-labeled preference data introduces high annotation costs and poses inherent limits on scalability.*

Consequently, scalable reward modeling (Gao et al., 2022) serves as a promising direction for capturing complex, multi-faceted objectives, thereby facilitating more efficient policy optimization.

In the real world, humans rarely rely solely on environmental feedback for long-horizon learning. Instead, they transform such feedback into *richer intrinsic rewards*, such as motivation or curiosity. As shown in Figure 2, we refer to this process as the conversion from *fast-rewards* to *slow-rewards*: environmental feedback first elicits direct, immediate sensations (fast), which are then gradually transformed into higher-order, abstract rewards (slow) through *introspective processing* (Qu et al., 2024; Zhang

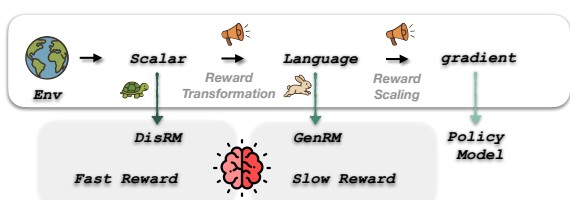

Figure 2: Illustrating the analogy between Dual-Reward method and the human learning process.

et al., 2025a). Motivated by this insight, we propose a **Dual-Reward** method: Discriminative Reward Models (DisRMs) (Liu et al., 2024; Yang et al., 2024b; Lou et al., 2025; Winata et al., 2025) captures *fast-rewards*, while Generative Reward Models (GenRMs) (Cao et al., 2024b; McAleese et al., 2024; Ye et al., 2024b; Liang et al., 2025; Liu et al., 2025) learns *slow-rewards*. The GenRM is designed to generate dynamics, fine-grained intrinsic signals based on the scalar output of the DisRM, thereby facilitating more effective policy optimization. Interestingly, from a modeling perspective, capturing fast and slow rewards exhibits an inverted relationship: DisRMs captures environmental feedback more slowly than GenRMs processes scalar rewards. This highlights *reward scaling* as a key feature: DisRMs gradually transforms rewards via the BT loss, while GenRMs leverages Test-time Scaling (TTS) (Snell et al., 2024; Uesato et al., 2022; Gulcehre et al., 2023) to extend its capacity and generalize across scenarios.

However, the step-by-step reasoning of the Dual-RMs is susceptible to compounding bias. That is, the training signals of GenRM cannot be effectively supervised, meaning that *false positives* may arise even when the predictions appear correct. To address this issue, we introduce a **meta-reward learning** algorithm, which designs a *unified intrinsic reward* (i.e., the meta-reward function) to enforce consistency among the DisRM, GenRM, and policy models. Specifically, we filter high-quality data from multiple rollout trajectories: GenRMs produce judge rationales for DisRM's results, the policy refines the responses, and DisRM re-evaluates the improved outputs. Meta-reward learning performs *general and automatic reward shaping* (Zou et al., 2019; Fu et al., 2025) for GenRMs by providing more precise training signals, thereby mitigating spurious correlations.

We present GRM-OMNI, an omni-modal reward model built on a meta-reward learning algorithm. GRM-OMNI employs a *criteria-based TTS* approach to uncover the underlying rationale, including both *criteria* and *judgments*, for aligning with preference labels or DisRM outputs. It can self-evolve by identifying the most critical criteria and optimizing their combinations to produce more accurate evaluations (Wang et al., 2024c). Experimental results demonstrate that GRM-OMNI achieves strong cross-modal transfer, yielding substantial gains on multimodal reward benchmarks while re-

lying solely on alignment in the language modality. Furthermore, continued training with additional multimodal data consistently enhances performance, markedly increasing the model's adaptability to diverse and complex scenarios.

> To highlight the main contributions and take-aways of our work:
>
> ➜ Conceptually, we demonstrate that **dual-reward design** and **meta-reward learning** can effectively scale reward modeling by bridging the gap between environment feedback and policy learning. In addition, our meta-reward function shapes rewards across both scalar and language signals, effectively reducing compounding bias.
>
> ➜ Methodologically, we propose a **criteria-based TTS** that improves the model's capacity to identify the most critical decision rationales via exploration and exploitation of criteria, thereby aligning DisRMs, GenRMs, and the Policy model through consistent criterion-based judgments.
>
> ➜ Empirically, we show that GRM-OMNI, trained only on language-based preference data, performs competitively on multi-modal benchmarks, highlighting the strong transferability of meta-reward learning to unseen scenarios.

## 2 META REWARD DESIGN

Meta-reward refers a *higher-order signal* that does not directly supervise policy behavior but instead evaluates the quality of a reward function itself. Designing a scalable meta-reward (Wu et al., 2024) that can generalize across scenarios or tasks remains a non-trivial challenge. Until now, there has been no widely accepted automated method for providing higher-order rewards to reward models. This challenge is difficult even for human annotators: real-world rewards are typically the result of multi-dimensional judgments (Li et al., 2023) (e.g., correctness, safety, fairness), and these dimensions often lack a unified scale (Wang et al., 2023; 2024b).

We now review the characteristics of traditional reward modeling approaches: DisRM learns pairwise preference annotations (easily accessible) during training and subsequently predict potential ranking scores for candidate items at test time. However, it remains sensitive to dataset biases and is limited by the *compression* of multi-dimensional judgments into a single scalar, complicating both interpretation and attribution. Additionally, DisRM cannot extend its capabilities by test-time scaling, as it is constrained by the size of available preference data. In contrast, GenRM employs LLMs to generate fine-grained rationales, providing a flexible mechanism to represent multi-dimensional objectives and underspecified constraints (Guo et al., 2025b; Wang et al., 2025a). But, the accuracy of these generated rationales is often difficult to verify, as hallucinations, spurious correlations, or biases may be amplified rather than mitigated. Consequently, DisRM and GenRM offer complementary strengths: DisRM provides ranking capabilities, while GenRM contributes language-based reasoning and potential for cross-task generalization, suggesting that hybrid approaches (Ankner et al., 2024; Yu et al., 2025) represent a promising direction.

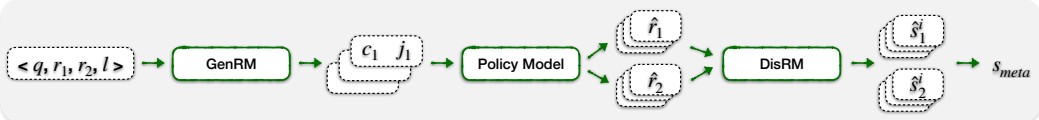

Figure 3: Overview of the proposed meta-reward score computation pipeline.

In this section, we propose a new *meta-reward function* that provides an ***intrinsic reward***, thereby enhancing the scalability of RMs. A key design principle of this meta-reward function is to ensure substantial differentiation between alternative reasoning trajectories (Liang et al., 2025), allowing the model to assign smooth, informative scores across a wide range of candidate outputs. By providing such finely graded signals, the meta-reward function not only facilitates effective ranking of trajectories but also supports the reward model in capturing subtle differences in reasoning quality, which is essential for multi-objective and fine-grained preference modeling. Given paired preference data $\langle q, r_1, r_2, l \rangle$ ($l \in 0, 1$), the **GenRM** generates multiple candidate criteria $\mathbf{c}_i$ together with

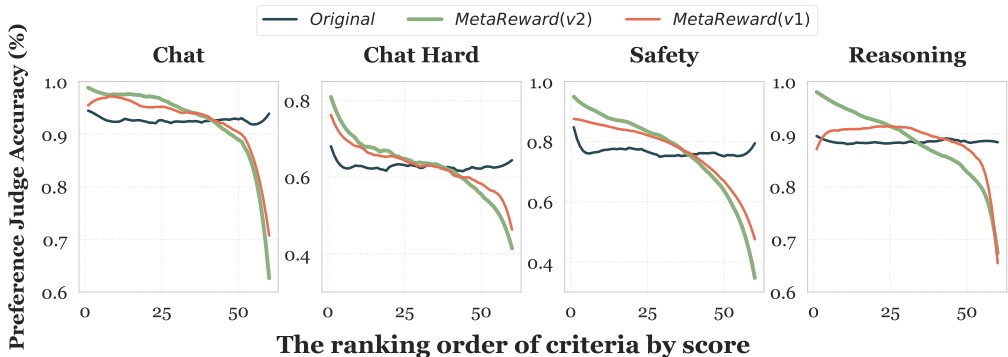

Figure 4: Accuracy curves of preference judgments for different criteria after ranking by $s_{\text{meta}}$ scores. "Original" denotes the results without $s_{\text{meta}}$ ranking, while "Meta-Reward (v1)" and "Meta-Reward (v2)" correspond to variants using different DisRM, with v2 demonstrating superior performance. For additional details and comparisons, please refer to the Appendix B.3.

corresponding judgments $\mathbf{j}_i$. These judgments serve as potential rationales for why the original responses were chosen or rejected. The **Policy model** then generates refined responses $\hat{r}_1^i$ and $\hat{r}_2^i$ corresponding to each candidate criterion $\mathbf{c}_i$ and judgment $\mathbf{j}_i$. Finally, the **DisRM** assigns scalar scores $s_1$ and $s_2$ to the original responses, and $\hat{s}_1^i$ and $\hat{s}_2^i$ to the refined responses; these scores are subsequently used to define the *meta-reward function*:

$$
s_{\text{meta}}(\tau_i) = \begin{cases} \hat{s}_2^i - s_2, & \text{if } j_i = 0 \text{ and } \hat{s}_1^i > \hat{s}_2^i, \\ \hat{s}_1^i - \hat{s}_2^i, & \text{if } j_i = 0 \text{ and } \hat{s}_1^i < \hat{s}_2^i, \\ \hat{s}_1^i - s_1, & \text{if } j_i = 1 \text{ and } \hat{s}_1^i < \hat{s}_2^i, \\ \hat{s}_2^i - \hat{s}_1^i, & \text{if } j_i = 1 \text{ and } \hat{s}_1^i > \hat{s}_2^i. \end{cases} \tag{1}
$$

Specifically, the reward model can use $s_{\text{meta}}$ to sample and prioritize multiple reasoning trajectories $\tau_i = (c_i, j_i, \hat{r}_1^i, \hat{r}_2^i, \hat{s}_1^i, \hat{s}_2^i)$, ensuring *consistency* and *causality*: In the first case, when the judgments from DisRM and GenRM are consistent (if $j_i = 0$ and $\hat{s}_1^i > \hat{s}_2^i$), $s_{\text{meta}} = \hat{s}_2^i - s_2$ measures the *improvement* of criterion $c_i$ on the rejected sample $s_2$. In the second case, when predictions are inconsistent, $s_{\text{meta}}$ reflects the difficulty of distinguishing between $\hat{r}_1^i$ and $\hat{r}_2^i$. Since the value of $\hat{s}_1^i - \hat{s}_2^i$ is negative, a higher meta-reward implies a larger gap. Moreover, if the policy model generates responses that receive high evaluations from DisRM under a given criterion, that criterion is assigned higher priority. This ensures that high-priority criteria are not arbitrary but instead causally linked to the policy's improvement, thereby preserving causality in the $s_{\text{meta}}$ calculation.

The proposed **meta-reward function** requires no additional training and can be seamlessly combined with any GenRM and DisRM. As shown in Figure 4, we evaluate its effectiveness on `RewardBench` using Qwen3-32B-AWQ as the policy model and GenRM, the Skywork-Reward model serving as DisRM. We observe that, after ranking by $s_{\text{meta}}$, the judge accuracy curves across different criteria become much smoother compared to the unsorted originals. This demonstrates that the **meta-reward** effectively captures and prioritizes the relative importance of different $c_i$.

> **Notes:** DisRM is designed to measure how GenRM generated criteria impact policy performance, thereby generating a ranking of criteria importance rather than selecting among GenRM outputs. The piecewise function is used to precisely capture these differences, enabling the reward function to produce smooth signals on the preference dataset.

## 3 GENERATIVE OMNI-MODALITY REWARD MODELING

### 3.1 META-REWARD LEARNING

Based on the meta-reward design, we propose a **meta-reward learning** algorithm that optimizes the reward function itself, improving the robustness of reward model training. Key features of the algorithm include:

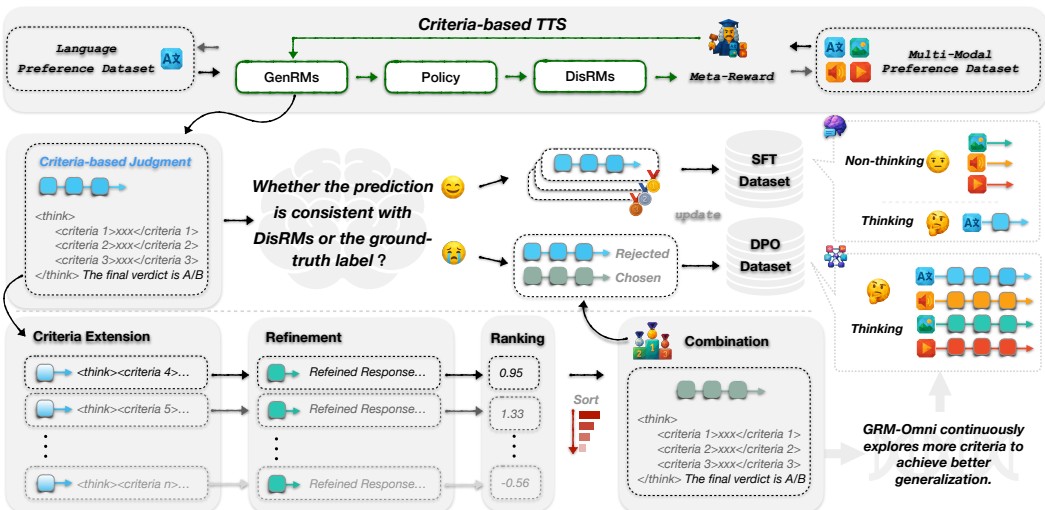

Figure 5: Overview of our meta-reward learning based DPO framework, which consists of two main stages: data synthesis and model training. In the SFT stage, the training corpus includes language-only data in the thinking mode and multimodal data in the non-thinking mode. During the DPO stage, thinking-style multimodal data can be *optionally* incorporated. Although the procedure can be iterated multiple times, we adopt a **single iteration** in practice for efficiency.

1. **Complex objectives**: Enables the reward model to generate multi-objective, fine-grained rationales underlying preference judgments (binary feedback).

2. **TTS support**: Supports test-time scaling, allowing the reward model to refine reward judgments using additional computational capacity without relying on extra annotations.

Our meta-reward learning is agnostic to existing RL algorithms, allowing flexible adoption depending on the task or environment. In this work, we implement a variant based on the Direct Preference Optimization (DPO) (Rafailov et al., 2023), which not only trains the model but also generates high-quality data for further optimization. As shown in Figure 5, we treat preference labels as a form of environmental feedback, and leverage the GenRMs, Policy Model, and DisRM to compute the meta-reward score. In our implementation, we employ open-source LLMs as both GenRMs and the policy model to synthesize preference data, while DisRM serves as the evaluator to collect high-quality training dataset.

## 3.2 CRITERIA-BASED TTS

**Trade-off between exploration and exploitation.** In traditional test-time scaling (TTS) (Zhang et al., 2025b; Ji et al., 2025), two conditions are typically required: 1) the policy must exhibit sufficient diversity, and 2) the verifier must provide reliable scoring. While our proposed meta-reward $s_{\text{meta}}$ addresses the ranking challenge, ensuring diverse reasoning trajectories remains non-trivial. Existing methods rely on sampling-based heuristics, which are constrained by the model's intrinsic capacity and thus fail to guarantee sufficient diversity. To address this, we introduce a **criteria-based TTS** strategy that leverages distinct criteria to guide the generation of diverse reasoning chains. Moreover, this approach naturally supports multi-objective and fine-grained preference modeling by tracing back decisions to their underlying atomic factors (criteria).

We interpret the optimization process of criteria-based TTS as a balance between *exploration* and *exploitation*. In the exploitation phase, GenRMs generate multiple candidate trajectories, which are ranked by $s_{\text{meta}}$ to select high-quality and diverse reasoning rationales. In the exploration phase, GenRMs perform *criteria extension*, synthesizing new decision dimensions from existing criteria. These extended criteria are then re-ranked and recombined using $s_{\text{meta}}$, allowing the model to refine its judgments by composing the most informative decision factors (Coste et al., 2024). The degree of exploration can be explicitly controlled by tuning the number of generated criteria, and in practice

it may be further adjusted based on the nature of the task. As shown in Appendix B.3, incorporating additional criteria under more challenging tasks significantly improves optimization capacity.

**Data synthesis and training.** We first synthesize criteria and the corresponding reasoning trajectories, thereby incentivizing the full capabilities of the omni-modality model. Concretely, the GenRM generates $k$ criteria under the current policy and produces a judgment based on these criteria. If the judgment is incorrect, the model performs *criteria expansion* by generating $n$ new criteria, which are then used to refine the original response. [1] The refined responses are subsequently evaluated by the DisRM, and the resulting scores are used to re-rank the candidate trajectories. From the top-ranked trajectories, the three most effective criteria are combined to produce improved judgments. This approach leverages the fact that combining different criteria can lead to more accurate and reliable scores, as demonstrated in prior work (Saha et al., 2024). The resulting improved judgments, together with outputs from the original policy, are stored as preference pairs in the DPO dataset. This design allows iterative updates of both the model and data, effectively accommodating the slow training dynamics of the DisRM.

### 3.3 GRM-OMNI

An important capability of criteria-based TTS is enabling the model to trace key decision criteria, which can intuitively generalize across different modalities. To verify this, we focus on a challenging multimodal reward modeling scenario: achieving cross-modal generalization using alignment and training solely in the language modality. A key challenge is that open-source multimodal backbone models are generally less capable than their language counterparts, particularly in instruction-following. Consequently, generating reward data with multimodal models is non-trivial. We adopt QWEN-OMNI (Xu et al., 2025) as the backbone and incorporate multimodal data during SFT training to expose the model to diverse input modalities. Specifically, multimodal inputs are paired with direct reasoning prompts, while language-only inputs employ CoT-based reasoning prompts. This design allows the reward model to adapt to multimodal inputs during training while preserving its ability to perform CoT reasoning at inference. Furthermore, by leveraging our meta-reward learning framework, the model is explicitly guided to identify and prioritize critical decision criteria across different modalities. As a result, the trained reward model not only generalizes effectively to unseen modalities but also maintains high-quality reasoning trajectories, facilitating robust preference judgments in complex, multimodal scenarios.

## 4 EXPERIMENTS

To train the reward model, we first collected a large corpus of open-source data, from which only a subset was sampled for training to ensure the query diversity. During the SFT stage, we employed QWEN3-32B-AWQ (Yang et al., 2025) to synthesize **22K SFT training examples**, while in the DPO stage, **20K DPO examples** were generated for model optimization. Although scaling up the dataset could further benefit model performance, exploring this scalability is beyond the scope of the current work. Detailed training hyperparameters and implementation are provided in Appendix B.2.

In our experiments, we conduct a comprehensive evaluation across multimodal reward modeling benchmarks, spanning two language benchmarks, four vision benchmarks, and two audio benchmarks, for a total of 22 task subsets (● `RewardBench`, ○ `RMB`, ■ `VL_RewardBench`, ❏ `Multimodal_RewardBench`, ▲ `GenAI_Bench (Image)`, ▼ `GenAI_Bench (Video)`, ◆ `Audio_Bench (Understanding)`, ❖ `Audio_Bench (Generation)`). These benchmarks span both understanding and generation tasks, with multimodal generation in particular requiring strong cross-modal reasoning capabilities. Therefore, this collection of benchmarks provides a comprehensive evaluation of the effectiveness of our proposed reward modeling approach. More detailed experimental settings and benchmark descriptions can be found in the Appendix B.1.

### 4.1 MAIN RESULTS

As shown in Table 1, we conduct a comprehensive evaluation of GRM-OMNI on eight multimodal benchmarks to examine its ability to generalize across diverse tasks and modalities. We consider

---

[1]In our experiments, we set $k$=3 and $n$=7.

Table 1: Performance of QWEN-OMNI across various benchmarks. Notably, on the Audio benchmark, QWEN-OMNI struggles with coherent reasoning and often produces repetitive outputs; therefore, we report results obtained by training on language-preference data. Additional results are provided in the Appendix B.4.

| Models | Size | Language | | Vision | | | | Audio | | Avg. |
|--------|------|---------|---------|---------|---------|---------|---------|---------|---------|------|
| | | ● | ○ | ■ | ❏ | ▲ | ▼ | ◆ | ❖ | |
| QWEN3-AWQ | 32B | 93.2 | 72.5 | - | - | - | - | - | - | - |
| QWEN-OMNI | 7B | 68.9 | 50.7 | 36.6 | 52.9 | 41.5 | 46.4 | 50.3 | 67.8 | 51.9 |
| GRM-OMNI-SFT | 7B | 82.2 | 64.5 | 53.3 | 58.5 | 63.6 | 70.6 | 58.0 | 70.0 | 65.1 |
| GRM-OMNI | 7B | 87.4 | 70.4 | 64.8 | 62.4 | 65.5 | 72.3 | 58.7 | 75.5 | 69.6 |

Table 2: Evaluation results on VL RewardBench benchmark.

| Methods | Inference | VL_RewardBench | | | Avg. |
|---------|-----------|------|--------|--------|------|
| | | Gen. | Hallu. | Reason. | |
| GPT-4o (2024-08-06) | *Thinking* | 49.1 | 67.6 | 70.5 | 62.4 |
| IXC-2.5-Reward | *Thinking* | 84.7 | 62.5 | 62.9 | 70.0 |
| UnifiedReward | *Thinking* | 76.5 | 58.1 | 65.1 | 66.6 |
| QWEN-OMNI | *Thinking* | 24.6 | 29.2 | 56.0 | 36.6 |
| GRM-OMNI | *Thinking* | 71.9 | 60.5 | 61.9 | 64.8 |
| GRM-OMNI ++ | *Thinking* | 72.0 | 69.8 | 62.3 | 68.0 |

two variants of our model: 1) GRM-OMNI-SFT, trained via SFT on criteria-synthesized data, and 2) GRM-OMNI, which further incorporates preference optimization through our meta-reward based DPO framework. When compared against the backbone model (QWEN-OMNI) and the synthesis-only model (QWEN3-AWQ), GRM-OMNI-SFT already achieves a significant improvement of 13.2%, highlighting the effectiveness of criteria-guided data synthesis in enhancing reward modeling capabilities. Beyond this, GRM-OMNI yields an additional 4.5% gain after preference optimization, demonstrating that meta-reward learning can further refine preference judgments. Interestingly, we observe that the improvements are consistent across nearly all benchmarks, with particularly strong gains in tasks requiring fine-grained multimodal reasoning. This suggests that the combination of criteria-based supervision and meta-reward guided optimization not only strengthens alignment in the language modality but also facilitates more robust cross-modal transfer.

## 4.2 CONTINUAL TRAINING WITH MULTIMODAL DATA

To further evaluate the generalization ability of our approach, we extend GRM-OMNI to the multimodal setting by incorporating multimodal policy and reward models for synthesizing new DPO datasets based on $D_{\text{R1-RM}}$ and $D_{\text{LLaVA}}$. Using the same scale of 20K preference pairs, we train the extended model, denoted as GRM-OMNI++. This design allows the model to adapt its reasoning and preference modeling capabilities beyond text, covering diverse input modalities.

As shown in Tables 2 and 3, GRM-OMNI++ consistently outperforms GPT-4o across multimodal benchmarks. Compared with models such as IXC-2.5 Reward and UnifiedReward, which rely on extensive amounts of multimodal training data, our approach benefits from the strong initialization provided by GRM-OMNI. This initialization substantially reduces the reliance on large-scale multimodal supervision, allowing the training process to focus more effectively on preference alignment rather than data scaling. These results highlight that our method not only delivers strong performance in language-only settings, but also scales efficiently to multimodal contexts, achieving competitive or superior results to state-of-the-art proprietary systems with significantly less multimodal data.

## 4.3 EXPLORATION CAPABILITIES

We further investigate whether training the reward model from scratch can yield more informative criteria, thereby extending its preference judgment capabilities. The motivation is to examine

Table 3: Evaluation results on Multimodal_RewardBench benchmark.

| Methods | Inference | Multimodal_RewardBench | | | | | | | Avg. |
| | | Corr. | Pref. | Know. | Math | Cod. | Safety | VQA | |
|---|---|---|---|---|---|---|---|---|---|
| GPT-4o | *Thinking* | 62.6 | 69.0 | 72.0 | 67.6 | 62.1 | 74.8 | 87.2 | 70.8 |
| Qwen2.5-VL-7B | *Thinking* | 58.1 | 61.0 | 56.8 | 54.7 | 47.9 | 56.1 | 68.7 | 57.6 |
| MM-RLHF-Reward | *Thinking* | 61.7 | 67.5 | 54.3 | 58.4 | 57.9 | 92.9 | 76.8 | 67.1 |
| GRM-OMNI | *Thinking* | 70.8 | 67.4 | 61.6 | 62.4 | 56.7 | 38.4 | 79.8 | 62.4 |
| GRM-OMNI ++ | *Thinking* | 72.2 | 67.7 | 67.1 | 71.2 | 58.8 | 88.1 | 84.5 | 72.8 |

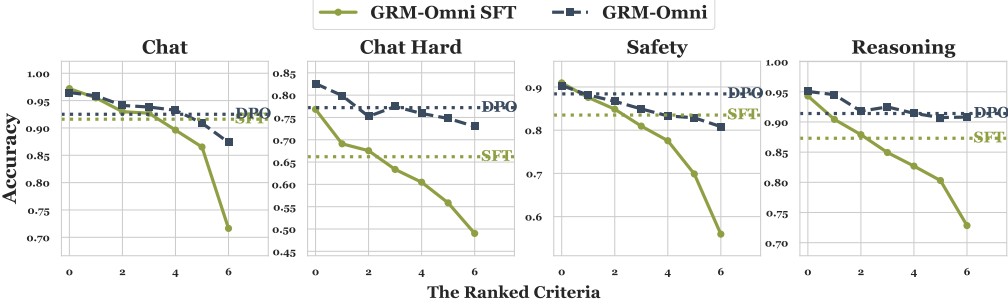

Figure 6: Exploration capabilities of GRM-OMNI on `RewardBench` after SFT and DPO training.

whether post-training allows the discovery of criteria that are both more generalizable and discriminative compared to those derived from strong foundation models. To ensure a fair comparison, we maintain the same policy model and DisRM as in the previous experiments.

As illustrated in Figure 6, we compare the performance of GRM-OMNI on `RewardBench` after SFT and DPO training, following criteria extension. Our results demonstrate that applying meta-reward guided ranking consistently improves the model's ability to explore informative trajectories. Interestingly, DPO models exhibit a higher lower bound compared to SFT models, suggesting that reinforcement learning can substantially enhance performance on challenging or low-quality samples, thereby improving the overall robustness of the reward model.

## 4.4 IMPACT ON DOWNSTREAM TASK PERFORMANCE

To further assess whether GRM-OMNI can enhance downstream task performance, we evaluate it on the Preference Proxy Evaluations (PPE) benchmark, which measures reward models via proxy tasks. We focus on three representative datasets from PPE, including MATH, to provide a comprehensive evaluation of model performance. For comparison, we consider strong baselines: Deepseek-GRM-27B that trained via GRPO, and Skywork-Reward-Llama-3.1-8B, which achieves high scores on Reward-Bench. As shown in Figure 7, GRM-OMNI consistently outperforms these baselines, demonstrating that our meta-reward

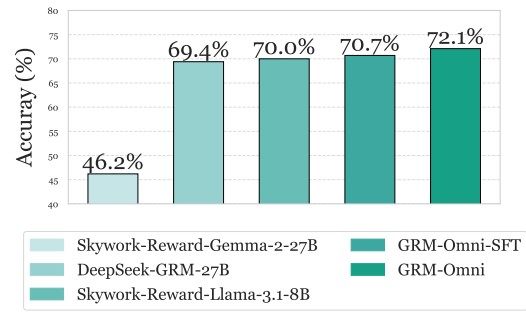

Figure 7: Evaluation of model performance on MATH benchmarks using the correctness accuracy metric.

learning approach enhances reward modeling and improves downstream task performance.

## 4.5 CASE STUDIES AND ERROR ANALYSIS

To better understand the strengths and limitations of GRM-OMNI, we perform case studies across multiple benchmarks, analyzing both successful and failed examples. Our goal is to identify cases where the model demonstrates clear improvements and to uncover common error patterns. For instance, in language-based dialogue tasks, given a query asking for an explanation of photosynthesis,

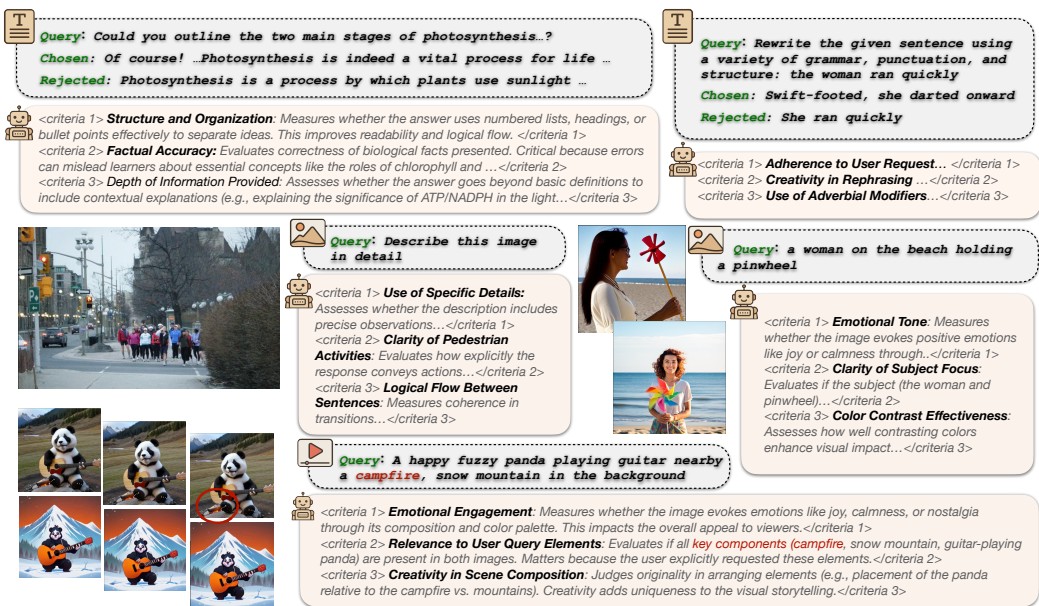

Figure 8: Examples of synthesized criteria generated by GRM-OMNI across multiple multimodal benchmarks, including language and visual dialogue, image generation, and video generation tasks.

the model identifies content **organization and structure** as the most critical criterion, followed by factual accuracy and content depth. When multiple responses are factually correct, what distinguishes them is primarily how well the content is organized. In visual dialogue tasks, the reward model emphasizes the **use of specific details** to distinguish between accepted and rejected responses. For video generation, the model emphasizes relevance to key query elements (e.g., a campfire) to accurately distinguish correct outputs from incorrect ones.

We further categorize the observed error types to better understand model limitations and guide future works. Across multiple benchmarks, we find that approximately 60% of failures stem from **biased or misaligned evaluation criteria**, 20% from **hallucinations or factual errors**, and 10% from **misinterpretation or incorrect application of criteria**. The remaining errors are often attributable to limitations in the underlying foundation models or inconsistencies in the benchmark itself. For example, in an image task asking the model to describe skiing scenes, the generated criteria focused on the evaluation team's composition and safety equipment rather than the skiing content itself, resulting in incorrect reward judgments. Similarly, in a kite-counting task, the model failed to identify the correct number of kites, reflecting inherent limitations in the backbone model's perception capabilities. These cases highlight that, although meta-reward learning enhances preference modeling, the quality and capacity of the underlying backbone model remain limiting factors. In future work, we aim to extend this approach to larger and more capable foundation models to further reduce such errors and improve robustness across diverse multimodal scenarios.

## 5 CONCLUSIONS

In this work, we identify a critical mismatch between reward modeling and policy optimization in the standard RLHF, and introduce meta-reward learning as a potential solution. We design a novel meta-reward function that not only captures fine-grained preference judgments but also provides a robust ordering mechanism to guide preference alignment. Building on this design, we further propose meta-reward learning variants of SFT and DPO, and incorporate a criteria-driven test-time scaling strategy to better explore diverse alignment objectives. Empirically, we demonstrate that GRM-OMNI, trained exclusively on textual data, generalizes remarkably well across eight multimodal benchmarks, substantially reducing reliance on large-scale preference data. These findings highlight the contribution of meta-reward learning: 1) as a more reliable paradigm for reward modeling, and 2) as a training process that inspires new directions for aligning future foundation models.

## ETHICS STATEMENT

This work complies with the ICLR Code of Ethics. All datasets used (as shown in Table 4) are publicly available and were utilized in accordance with their licenses. All models employed in our work (e.g., *Qwen3-32B-AWQ*,*Skywork-Reward V2*) are open-source and accessible under their respective usage terms. No personally identifiable or sensitive information was included. Our experiments did not involve human subjects or animal testing. We took care to minimize potential risks such as unfair bias or misuse, and our methods are designed to promote reproducibility and transparency.

## REPRODUCIBILITY STATEMENT

We have made every effort to ensure that the results presented in this paper are reproducible. All datasets used for training and evaluation are publicly available, and the corresponding access links are provided in Table 4. Details of our training procedure, including hyperparameters, optimization settings, and hardware configurations, are documented in the Appendix B. To further facilitate reproducibility, we will release all code, synthetic training data, and model checkpoints upon publication.

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

**The appendix includes detailed experimental setups and corresponding results, along with a detailed exhibition of the reward modeling methods employed for different modalities.**

## A  RELATED WORK

Aligning LLMs with human intent (Bai et al., 2022; Christiano et al., 2017; Ouyang et al., 2022) has emerged as a core challenge in the pursuit of AGI. This challenge spans both language-only and multimodal reasoning tasks. Unlike *imitation learning*, which is typically implemented via supervised fine-tuning (SFT), RL-based approaches (Schulman et al., 2017; Shao et al., 2024) leverage an auxiliary reward model to provide richer supervision signals. These reward models enable LLMs not only to generate high-quality reasoning trajectories, but also to perform fine-grained preference modeling that supports the optimization of underspecified or ambiguous objectives (Chen et al., 2025b; Chu et al., 2025). In this section, we provide a comprehensive review of existing reward modeling paradigms, including traditional **DisRMs** and **GenRMs**. DisRMs as a "fast-thinking" mechanism, providing scalar scores to guide model behavior. In contrast, GenRMs run a "slow-thinking", generating explicit CoT that capture the underlying rationales behind judgments.

While reward modeling can take various forms: such as *point-wise*, *pair-wise*, or *list-wise* ranking, this work focuses on the **pairwise**, which remains the most widely adopted approach in RLHF. Pairwise reward modeling aims to compare model outputs and identify which response better aligns with human preferences, thereby enabling the construction of reward functions to guide model optimization. Formally, let $x$ denote a user query, and $y^+$, $y^-$ denote two responses generated by the LLMs, where $y^+$ is preferred over $y^-$ by a human annotator. The reward model $r_\theta(x, y)$ is parameterized as a function that evaluates the quality of a given paired data. The learning objective of RMs is to ensure that:

$$r_\theta(x, y^+) > r_\theta(x, y^-).$$

As illustrated in Figure 9, both Scalar RMs and GenRMs can optimize this pairwise objective. Scalar RMs project contextual representations into scalar scores via an MLP head and are typically trained with the Bradley–Terry loss. In contrast, GenRMs treat reward modeling as a conditional generation task and are optimized using standard language modeling loss functions.

### A.1  DISCRIMNATIVE REWARD MODELING

The Bradley-Terry (BT) model is a classical probabilistic framework for modeling preferences based on pairwise comparisons. It assumes that each option is assigned a utility score, and the probabil-

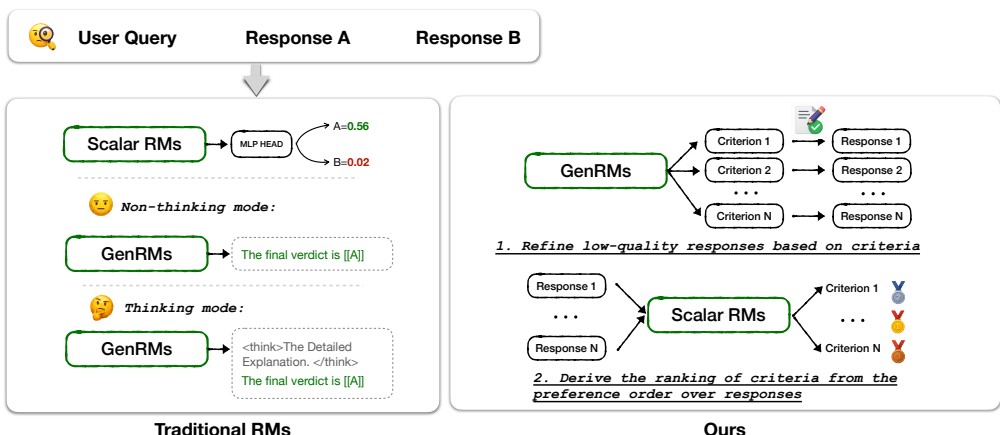

Figure 9: Illustration of the differences between GenRM and DisRM, and how our approach combines their complementary strengths.

ity of one option being preferred over another depends on the relative magnitude of their scores. Formally, for a pair of options i and j, the preference probability is given by:

$$P(i \succ j) = \frac{e^{r(i)}}{e^{r(i)} + e^{r(j)}} \tag{2}$$

where $r(i)$ and $r(j)$ denote the scores (log-utilities) of options $i$ and $j$, respectively. This is equivalent to applying a softmax function over the two scores. A notable application is in the Chatbot Arena [2], where outputs from different models are evaluated through pairwise comparisons, and these comparisons are aggregated to produce a global preference ranking or implicit reward signal Sun et al. (2025).

In several recent works, Yuan et al. (2024) extend the Bradley–Terry (BT) framework by incorporating *absolute rewards* for individual actions, improving its suitability for binary comparison tasks. Yang et al. (2024a) impose regularization on the internal representations of reward models, enhancing their generalization to out-of-distribution (OOD) examples and mitigating overfitting to specific training distributions. Additionally, multi-objective reward formulations have been proposed to capture diverse and nuanced human preferences, allowing models to reason over trade-offs across multiple criteria (Wang et al., 2024a). However, these methods typically do not scale well, because they depend on large amounts of labeled data and struggle to speed up the model's learning and improvement.

## A.2 GENERATIVE REWARD MODELING

The emergence of GenRMs has been largely enabled by advances in LLMs (Zheng et al., 2023; Mahan et al., 2024), particularly their ability to leverage self-improvement techniques such as CoT reasoning and TTS. Recent work (Cao et al., 2024a; Ye et al., 2024a) has explored GenRMs that optimize reward models through preference modeling objectives, including both pairwise and single-point rewards. Beyond scoring, GenRMs can serve as feedback mechanisms or even assist in correcting errors in tasks such as mathematics (Gao et al., 2024; Zhang et al., 2024). A key advantage of GenRMs is their ability to produce interpretable reasoning trajectories, which can guide humans or downstream models for further refinement.

Generative reward models fall into two broad categories: *Thinking* and *Non-thinking*, depending on whether they explicitly produce CoT trajectories. Non-thinking models use CoT during training, and their performance closely matches DisRMs in practice. In contrast, Thinking models rely on high-quality CoTs and are susceptible to error accumulation due to they generate intermediate reasoning steps, which can limit their effectiveness. Yet this very capability also enables TTS, un-

[2]https://lmarena.ai/leaderboard

Table 4: Overview of the collected multimodal open-source preference dataset used for training.

| Modality | Task | Dataset | Number (K) | Huggingface |
|---|---|---|---|---|
| **Language** | *U&G* | $\mathcal{D}_{\text{Sky}}$ | 77 | Skywork/Skywork-Reward-Preference-80K-v0.2 [3] |
| **Image** | *Understanding* | $\mathcal{D}_{\text{R1-RM}}$ | 17.3 | yifanzhang114/R1-Reward-RL [4] |
| | | $\mathcal{D}_{\text{LLaVA}}$ | 34.7 | CodeGoat24/LLaVA-Critic-113k |
| | *Generation* | $\mathcal{D}_{\text{HPD}}$ | 24.0 | CodeGoat24/HPD [5] |
| | | $\mathcal{D}_{\text{OIP}}$ | 7.4 | CodeGoat24/OIP [6] |
| | | $\mathcal{D}_{\text{EvalMuse}}$ | 3.0 | CodeGoat24/EvalMuse [7] |
| **Video** | *Understanding* | $\mathcal{D}_{\text{ShareGPTVideo}}$ | 26.8 | CodeGoat24/ShareGPTVideo-DPO [8] |
| | *Generation* | $\mathcal{D}_{\text{VideoDPO}}$ | 10.0 | CodeGoat24/VideoDPO [9] |
| | | $\mathcal{D}_{\text{T2VHP}}$ | 5.7 | CodeGoat24/Text-2-Video-Human-Preferences [10] |
| **Audio** | *Understanding* | $\mathcal{D}_{\text{Align-Any-TA2T}}$ | 30.5 | PKU-Alignment/align-anything [11] |
| | *Generation* | $\mathcal{D}_{\text{Align-Any-T2A}}$ | 10.9 | PKU-Alignment/align-anything |

Table 5: Evaluation benchmarks for assessing preference judgment capabilities.

| Modality | Benchmark | Task | Huggingface |
|---|---|---|---|
| **Language** | RewardBench | *U&G* | allenai/reward-bench [12] |
| | RMB | *U&G* | RMB-Reward-Model-Benchmark [13] |
| **Vison** | VL_RewardBench | *Uderstanding* | MMInstruction/VL-RewardBench [14] |
| | Mutilmodal RewardBench | *Uderstanding* | multimodal_rewardbench [15] |
| | GenAI-Bench-Image | *Generation* | TIGER-Lab/GenAI-Bench [16] |
| | GenAI-Bench-Video | *Generation* | TIGER-Lab/GenAI-Bench [17] |
| **Audio** | Align_Anything_TA2T | *Uderstanding* | PKU-Alignment/align-anything [18] |
| | Align_Anything_T2A | *Generation* | PKU-Alignment/align-anything |

locking reasoning capabilities and potential. Our method improves the fidelity of these training CoTs, strengthening the model's ability to make accurate and interpretable reward judgments.

# B EXPERIMENTS DETAILS

## B.1 DATASETS

To train the reward model, we first aggregated a large corpus from open-source datasets, all of which are publicly available with download links provided. Table 4 summarizes the final dataset statistics. Rather than using the entire corpus, we **randomly sampled subsets** to ensure broad coverage of the query distribution, promoting robustness in the training process. During the training of GRM-OMNI, we only rely on language data to synthesize criteria and judgements. The role of other multimodal data was limited to enabling the base model to handle inputs from multiple modalities. We achieved this through a prompt-switching strategy: For language training, we used language

---

[3] https://huggingface.co/datasets/Skywork/Skywork-Reward-Preference-80K-v0.2

[4] https://huggingface.co/datasets/yifanzhang114/R1-Reward-RL

[5] https://huggingface.co/datasets/CodeGoat24/HPD

[6] https://huggingface.co/datasets/CodeGoat24/OIP

[7] https://huggingface.co/datasets/CodeGoat24/EvalMuse

[8] https://huggingface.co/datasets/CodeGoat24/ShareGPTVideo-DPO

[9] https://huggingface.co/datasets/CodeGoat24/VideoDPO

[10] https://huggingface.co/datasets/CodeGoat24/Text-2-Video-Human-Preferences

[11] https://huggingface.co/datasets/PKU-Alignment/align-anything

[12] https://huggingface.co/datasets/allenai/reward-bench

[13] https://github.com/Zhou-Zoey/RMB-Reward-Model-Benchmark

[14] https://huggingface.co/datasets/MMInstruction/VL-RewardBench

[15] https://github.com/facebookresearch/multimodal_rewardbench

[16] https://huggingface.co/datasets/TIGER-Lab/GenAI-Bench

[17] https://huggingface.co/datasets/TIGER-Lab/GenAI-Bench

[18] https://huggingface.co/datasets/PKU-Alignment/align-anything

Table 6: Hyperparameters used for training GRM-OMNI.

| Hyperparameter | Value |
|---|---|
| Number of GPUs | 8 |
| Per device train batch size | 2 |
| Gradient accumulation steps | 8 |
| Sequence cutoff length | 4096 |
| Number of training epochs | 1 |
| Learning rate (SFT) | 1e-5 |
| Learning rate (DPO) | 1e-6 |
| Learning rate scheduler | cosine |
| Warmup ratio | 0.1 |

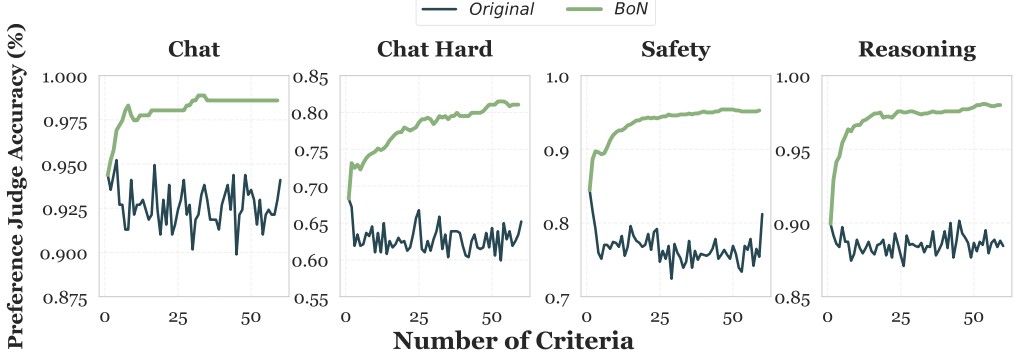

Figure 10: Best-of-N (BoN) evaluation of criteria after ranking

prompts paired with thinking-style CoT responses. For multimodal training, we used multimodal prompts paired with non-thinking responses. After this stage, the trained model can adaptively switch prompts to produce CoT reasoning in multimodal scenarios as well.

For evaluation, we benchmark GRM-Omni across multiple modalities. Specifically, we consider `RewardBench` and `RMB` for language understanding, `VL RewardBench` and `Multimodal RewardBench` for vision understanding, `GenAI Bench` for vision generation, and `Align Anything` (referred to here as the `Audio Bench`) for video and audio. Table 5 provides a summary of the dataset statistics. We adopt accuracy as the evaluation metric across all benchmarks to consistently measure preference judgment performance.

## B.2 TRAINING AND INFERENCE

During data synthesis, we adopt Qwen3-32B-AWQ as the GenRM and Skywork-Reward V2 as the DisRM to construct training data for GRM-LANG. In the model sampling process, we further explore 7 new criteria to obtain higher-quality results. For multimodal reward modeling, we employ Qwen2.5-Omni-7B as the backbone model. To generate CoT data for multimodal training, we use the trained GRM-LANG as GenRM and Skywork-VL-Reward for data selection. As shown in Table 6, we report the hyperparameters used in the SFT and DPO stages to facilitate reproducibility.

## B.3 EMPIRICAL STUDIES ON META-REWARD DESIGN

To better understand how criteria relate to the meta-reward function, we conduct an in-depth analysis focusing on the following questions:

1. **Scalability**: Do preference judgments scale as the number of criteria increases?

2. **Voting benefit**: Can ranked criteria still yield improvements when aggregated via voting?

3. **Score distribution**: What is the value range of the meta-reward function?

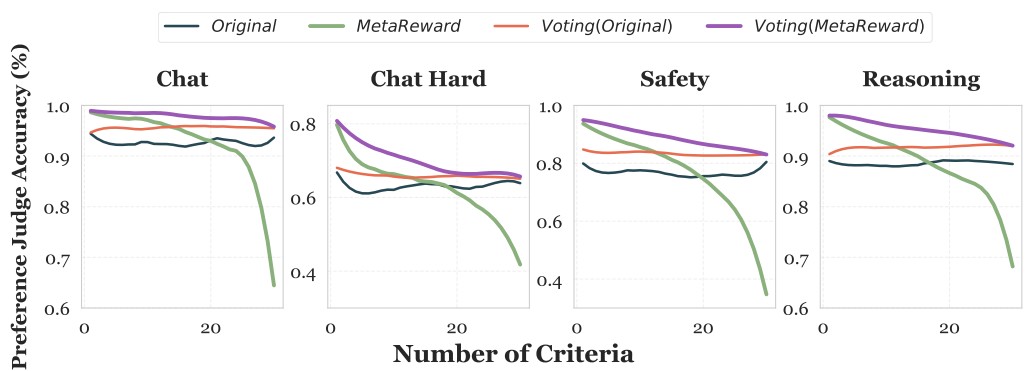

Figure 11: Impact of voting on the ranking of criteria.

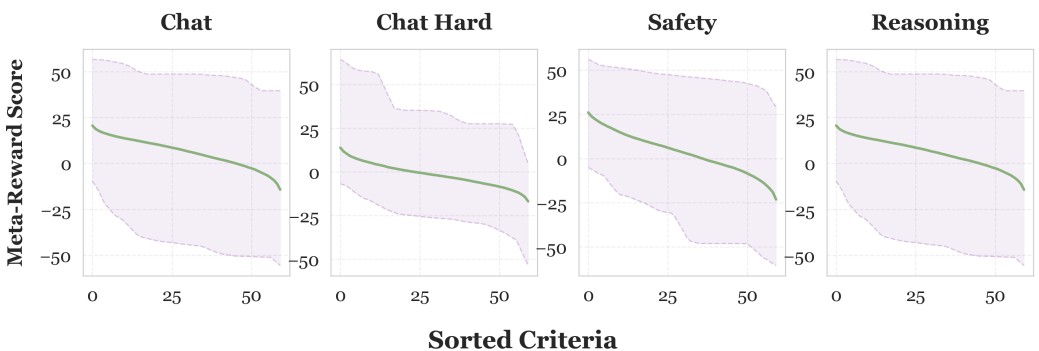

Figure 12: The ranges of meta-reward scores across different subsets.

All experiments were conducted on **RewardBench**. For the GenRM and Policy Model, we used Qwen-3-3B-AWQ, while DisRM was instantiated with Skywork-Reward-V2 [19] (with v1 [20] referring to Skywork-Reward-V1).

First, to examine the impact of the number of criteria on preference judgment accuracy, we compared the performance of the original 60 criteria without ranking against the performance after ranking, measured via Bag-of-N (BoN). As shown in Figure 10, BoN values increase as the number of criteria grows, suggesting that incorporating more criteria enhances the model's overall performance. This effect is especially evident on the **Chat Hard** subset, where the BoN curve exhibits a consistent upward trend. These results indicate that more difficult examples benefit from a larger set of criteria to arrive at correct judgments. [21]

We also investigate whether *voting* can enhance preference judgment accuracy after criteria ranking. As shown in Figure 11, *voting* improves judgment correctness for unsorted criteria. In contrast, once the criteria are ranked, applying large-scale *voting* offers minimal additional benefit. This suggests that the ranking process already identifies the most informative criteria for accurate judgments.

We further analyze the range of meta-reward scores by examining their mean, maximum, and minimum values (Figure 12). The results indicate that meta-reward scores can be either positive or negative, highlighting the necessity of handling reward signals in a segmented manner. Furthermore, the mean score curves provide a reliable signal for comparing and ranking different criteria.

---

[19] https://huggingface.co/Skywork/Skywork-Reward-V2-Llama-3.1-8B

[20] https://huggingface.co/Skywork/Skywork-Reward-Llama-3.1-8B

[21] Specifically, these examples of the **Chat Hard** subset correspond to the lowest-scoring and most challenging cases in **RewardBench**.

Table 7: Evaluation results on RewardBench benchmark.

| Methods | Inference | RewardBench | | | | Avg. |
|---------|-----------|------|---------|--------|-----------|------|
| | | Chat | Chat H. | Safety | Reasoning | |
| *DisRMs* | | | | | | |
| **ArmoRM-Llama3-8B** | *Non-Thinking* | 96.9 | 76.8 | 90.5 | 97.3 | 90.4 |
| **Skywork-Llama-8B** | *Non-Thinking* | 95.8 | 87.3 | 90.8 | 96.2 | 92.5 |
| **Skywork-Gemma-27B** | *Non-Thinking* | 95.8 | 91.4 | 91.9 | 96.1 | 93.8 |
| *GenRMs* | | | | | | |
| **GPT-4o** | *Thinking* | 96.1 | 76.1 | 88.1 | 86.6 | 86.7 |
| **Gemini-1.5-pro** | *Thinking* | 92.3 | 80.6 | 87.9 | 92.0 | 88.2 |
| **DeepSeek-R1** | *Thinking* | 93.6 | 79.2 | 86.9 | 97.4 | 89.3 |
| **DeepSeek-GRM-27B** | *Thinking* | 94.1 | 78.3 | 88.0 | 83.8 | 86.0 |
| **IXC-2.5-Reward** | *Thinking* | 90.8 | 83.8 | 87.8 | 90.0 | 88.6 |
| **RM-R1-Qwen-7B** | *Thinking* | 94.1 | 74.6 | 85.2 | 86.7 | 85.2 |
| **QWEN-OMNI** | *Thinking* | 85.8 | 48.7 | 75.3 | 65.4 | 68.9 |
| **GRM-OMNI-SFT** | *Thinking* | 91.6 | 66.2 | 83.5 | 87.3 | 82.2 |
| **GRM-OMNI** | *Thinking* | 92.5 | 77.2 | 88.4 | 91.4 | 87.4 |

Table 8: Evaluation results on RMB benchmark.

| Methods | Inference | RMB | | | | Avg. |
|---------|-----------|-----------|-----------|------------|------------|------|
| | | BoN Help. | BoN Harm. | Pair Help. | Pair Harm. | |
| *DisRMs* | | | | | | |
| **ArmoRM-Llama3-8B** | *Non-Thinking* | 63.6 | 49.7 | 78.7 | 66.3 | 64.6 |
| **DeepSeek-BTRM-27B** | *Non-Thinking* | 64.0 | 33.6 | 83.0 | 51.0 | 57.9 |
| **Skywork-Reward-Llama** | *Non-Thinking* | 62.7 | 60.3 | 78.1 | 75.9 | 69.3 |
| *GenRMs* | | | | | | |
| **CLoud-Gemma-27B** | *Thinking* | 64.7 | 41.7 | 81.1 | 66.1 | 63.4 |
| **DeepSeek-PairRM-27B** | *Thinking* | 59.9 | 34.1 | 83.3 | 55.5 | 58.2 |
| **DeepSeek-GRM-27B** | *Thinking* | 62.3 | 57.0 | 80.5 | 76.1 | 69.0 |
| **Qwen3-AWQ** | *Thinking* | 66.5 | 64.3 | 81.1 | 77.9 | 72.5 |
| **QWEN-OMNI** | *Thinking* | 41.1 | 40.0 | 62.9 | 58.8 | 50.7 |
| **GRM-OMNI-SFT** | *Thinking* | 58.9 | 52.1 | 75.8 | 71.0 | 64.5 |
| **GRM-OMNI** | *Thinking* | 60.6 | 64.4 | 77.8 | 78.6 | 70.4 |

## B.4 MAIN TABLE RESULTS

We conduct a comprehensive evaluation across multiple benchmarks, reporting results on each sub-dataset to better capture the strengths and limitations of different reward modeling approaches:

1. **RewardBench** As shown in Figure 7, we provide a direct comparison between discriminative and generative reward models. Specifically, the discriminative model includes *Skywork-Llama-8B* (Liu et al., 2024), while the generative model includes *RM-R1-Qwen-7B* (Chen et al., 2025a).

2. **RMB** As shown in Figure 8, we compare against state-of-the-art generative models including *DeepSeek-GRM-27B* (Liu et al., 2025).

3. **VL RewardBench** As shown in Figure 9,we evaluate multimodal reward models, such as *MM-RLHF-Reward* (Zhang et al., 2025c).

4. **Multimodal RewardBench** As shown in Figure 10, we compare with vision-language reward models *Qwen2.5-VL-7B* (Bai et al., 2025).

5. **GenAI Bench** As shown in Figure 11,we compare with the strong baseline *UnifiedReward* (Wang et al., 2025b).

6. **Audio Bench** As shown in Figure 12,we compare with Qwen-omni (Xu et al., 2025).

Table 9: Evaluation results on VL RewardBench benchmark.

| Methods | Inference | VL_RewardBench | | | Avg. |
|---------|-----------|------|--------|---------|------|
| | | Gen. | Hallu. | Reason. | |
| **GPT-4o (2024-08-06)** | *Thinking* | 49.1 | 67.6 | 70.5 | 62.4 |
| **Claude-3.7-Sonnet** | *Thinking* | 68.1 | 70.7 | 60.8 | 66.5 |
| **MM-RLHF-Reward** | *Thinking* | 45.0 | 50.5 | 57.6 | 51.0 |
| **IXC-2.5-Reward** | *Thinking* | 84.7 | 62.5 | 62.9 | 70.0 |
| **R1-Reward** | *Thinking* | 63.8 | 85.7 | 64.8 | 71.4 |
| **UnifiedReward-Think** | *Non-Thinking* | 77.9 | 70.5 | 65.4 | 71.3 |
| **UnifiedReward-Think** | *Thinking* | 78.1 | 72.7 | 66.0 | 72.3 |
| **QWEN-OMNI** | *Thinking* | 24.6 | 29.2 | 56.0 | 36.6 |
| **GRM-OMNI-SFT** | *Thinking* | 54.0 | 44.8 | 61.0 | 53.3 |
| **GRM-OMNI** | *Thinking* | 71.9 | 60.5 | 61.9 | 64.8 |

Table 10: Evaluation results on Multimodal_RewardBench benchmark.

| Methods | Inference | Multimodal_RewardBench | | | | | | | Avg. |
|---------|-----------|-------|-------|-------|------|------|--------|------|------|
| | | Corr. | Pref. | Know. | Math | Cod. | Safety | VQA | |
| **GPT-4o** | *Thinking* | 62.6 | 69.0 | 72.0 | 67.6 | 62.1 | 74.8 | 87.2 | 70.8 |
| **Qwen2.5-VL-7B** | *Thinking* | 58.1 | 61.0 | 56.8 | 54.7 | 47.9 | 56.1 | 68.7 | 57.6 |
| **MM-RLHF-Reward** | *Thinking* | 61.7 | 67.5 | 54.3 | 58.4 | 57.9 | 92.9 | 76.8 | 67.1 |
| **LLama3.2-11B** | *Thinking* | 57.8 | 65.8 | 55.5 | 51.2 | 51.2 | 35.5 | 55.8 | 52.4 |
| **QWEN-OMNI** | *Thinking* | 56.0 | 61.2 | 55.1 | 57.6 | 48.5 | 33.3 | 58.3 | 52.9 |
| **GRM-OMNI-SFT** | *Thinking* | 58.9 | 68.6 | 61.1 | 64.1 | 52.5 | 33.1 | 73.1 | 58.5 |
| **GRM-OMNI** | *Thinking* | 70.8 | 67.4 | 61.6 | 62.4 | 56.7 | 38.4 | 79.8 | 62.4 |

## C  LLM USAGE

LLMs were used to aid in the writing and polishing of this manuscript. Specifically, we used OpenAI's ChatGPT-4 to refine the wording of our paper, improve academic tone, and assist in figure preparation. All text generated by the LLM was carefully reviewed and verified by the authors to ensure accuracy, faithfulness to the research, and consistency with our scientific contributions. The LLM was not used to generate novel ideas, design experiments, or produce results; its role was limited to language editing and presentation support. The authors take full responsibility for the content of the manuscript, including any text generated or polished by the LLM. We have ensured that the LLM-generated text adheres to ethical guidelines and does not contribute to plagiarism or scientific misconduct.

## D  PROMPT ENGINEERING

### D.1  DATA SYNTHESIS

As shown in Figures 13, 14 and 15, here we detail the prompt template employed to guide our meta-reward learning framework in judging between two responses. The template explicitly instructs the model to compare candidate responses according to multiple evaluation criteria, including both standard dimensions and our self-synthesized criteria, ensuring that the judgment process is systematic and aligned with task-specific requirements.

### D.2  INFERENCE

The prompt formats used for different input modalities or task settings are shown in Figures 16, illustrating the instruction templates adopted during training and inference.

Table 11: Evaluation results on GenAI benchmark.

| Methods | Inference | GenAI_Bench | | Avg. |
| --- | --- | --- | --- | --- |
| | | Image Gen | Video Gen | |
| **ImageReward** | *Thinking* | 65.0 | 73.1 | 69.1 |
| **VisionReward** | *Thinking* | 66.4 | 73.3 | 69.9 |
| **UnifiedReward** | *Thinking* | 70.9 | 77.2 | 74.1 |
| **UnifiedReward-Think** | *Non-Thinking* | 71.9 | 81.6 | 76.8 |
| **UnifiedReward-Think** | *Thinking* | 72.5 | 82.3 | 77.4 |
| **QWEN-OMNI** | *Thinking* | 41.5 | 46.4 | 44.0 |
| **GRM-OMNI-SFT** | *Thinking* | 63.6 | 70.6 | 67.1 |
| **GRM-OMNI** | *Thinking* | 65.5 | 72.3 | 68.9 |

Table 12: Evaluation results on Audio benchmark.

| Methods | Inference | Audio_Bench | | Avg. |
| --- | --- | --- | --- | --- |
| | | Audio Und | Audio Gen | |
| **QWEN-OMNI-SKYWORK** | *Thinking* | 50.3 | 67.8 | 59.0 |
| **GRM-OMNI-SFT** | *Thinking* | 58.0 | 70.0 | 64.0 |
| **GRM-OMNI** | *Thinking* | 58.7 | 75.5 | 67.1 |

Figure 13: Prompt for Criteria Generation.

**Thinking Prompt**

You are an expert in generating evaluation criteria. Given a user query and two assistant replies (which may include Text, Image, Video, and Audio), your task is to create **exactly 10 evaluation criteria** that best distinguish the strengths and weaknesses of the two responses. ### Instructions:1. Carefully compare the two replies and identify meaningful differences in how they respond, these could involve reasoning, factual accuracy, structure, clarity, creativity, style, or multimodal use. 2. From those differences, derive 10 context-specific criteria. Each criterion should help a third-party evaluator compare the two replies fairly and systematically. 3. Ensure that none of your criteria overlap with the given candidate criteria. If no candidate criteria are provided, propose the 10 criteria you consider most important. 4. For each criterion, provide a short, clear name, followed by a brief explanation of what it evaluates and why it matters in this context. ### Output format: <criteria 1>**Name**: Explanation<criteria 1>...<criteria 10>**Name**: Explanation</criteria 10>### Input: [User Question]:{query} [The Start of Assistant A's Answer]:{response_1}[The End of Assistant A's Answer] [The Start of Assistant B's Answer]: {response_2}[The End of Assistant B's Answer] [The Candidate Criteria Start]: {candidate_criteria} [The Candidate Criteria End]

Figure 14: Prompt for Judgment Based on a Given Criterion.

**Thinking Prompt**

You are an expert evaluator. Your task is to assess how well each assistant response satisfies the user's query, strictly based on the provided [Evaluation Criteria]. ### Instructions: 1. Carefully read the user query, the two assistant responses, and the [Evaluation Criteria]. 2. For each response, provide a step-by-step analysis of how effectively it addresses the user's query in relation to the criteria. Support your evaluation with specific evidence from the response content. 3. Your judgment must be strict, fair, and explicitly grounded in the Evaluation Criteria. 4. Do not assume one response is better unless the evidence clearly shows it satisfies the query more effectively. ### Final Output Format: 1. Present your results in three sections using the following format: ### Final Output Format: <Judge A>[Analysis of Response A: Evaluate against each Evaluation Criteria. Be explicit about strengths and weaknesses. Clearly state where improvements are needed and explain why. Assign an overall score from 1–10.]</Judge A><Judge B>[Analysis of Response B: same requirements as above.]</Judge B># The Final Verdict is [[A]] or [[B]]. 2. Assign an overall score to each response (integer from 1 to 10): + 1 = Does not satisfy the query at all under the criteria + 6 = Partially satisfies the query + 10 = Fully satisfies the query with excellence across all criteria 3. At the end, output your analysis and then give the final decision in this exact format: If Assistant A is better: explanation followed by [[A]]; If Assistant B is better: explanation followed by [[B]]. Only output one of the tags ([[A]] or [[B]]) on the final answer line, and nothing else. ###Input [User Query]{query} [The Start of Assistant A's Answer]{response_1}[The End of Assistant A's Answer] [The Start of Assistant B's Answer]{response_2}[The End of Assistant B's Answer] [The Begin of Evaluation Criteria]{criteria}[The End of Evaluation Criteria]

Figure 15: Prompt for Refinement Based on Judgment.

**Thinking Prompt**

You are tasked with revising the assistant responses based on a user query and a provided critique. ### Instructions: 1. Make only objective and necessary edits that directly address the specific points raised in the critique. 2. Do **not** change any content that is not explicitly mentioned in the critique. 3. Do **not** introduce new ideas, rephrase unrelated sections, or make stylistic edits beyond what the critique specifies. 4. Your revisions must be accurate, minimal, and strictly aligned with the critique. ### Output Format: Return **only** the fully revised response. Do **not** include any explanations, comments, or metadata. ### Input [User Query] {query} [The Start of Responses] {response} [The End of Responses] [The Start of Critique]{judge} [The End of Critique] Please return revised responses:

Figure 16: Prompt for Direct Judgment.

> **Thinking Prompt**
>
> You are a fair, professional, and neutral multimodal AI evaluator. You are tasked with evaluating two different multimodal responses (Text, Image, Video, Audio) generated for the same user query, and determining which one is better. ### Instructions 1. Comparison Basis. Carefully compare the two responses and identify meaningful differences in how they address the user's query. Differences could involve: reasoning, factual accuracy, clarity, structure, creativity, or use of multimodal elements. 2. Criteria Creation. From these differences, derive exactly three evaluation criteria that best distinguish the strengths and weaknesses of the two responses. Each criterion should have a short name and a brief explanation of what it evaluates and why it matters. Ensure the criteria do not overlap with each other. 3. Evaluation. For each criterion, analyze Response A and Response B step by step. Support your evaluation with clear, specific evidence from the responses. Maintain a strict, fair, and grounded judgment. 4. Verdict. Your final decision must be '[[A]]' or '[[B]]' (no ties allowed). Base your verdict solely on which response performs better under the three criteria. ### Output Format<think><criteria 1>Name. Explanation. <Judge A>Analysis of A.</Judge A><Judge B>Analysis of B.</Judge B>Based on the criteria, the verdict is [[A]] or [[B]]</criteria 1>... </think>The final verdict is [[A]] or [[B]] ### Input:[User Question]: {query} [The Start of Assistant A's Answer]: {response_1} [The End of Assistant A's Answer] [The Start of Assistant B's Answer]: {response_2} [The End of Assistant B's Answer] Please output your analysis and final verdict:

