# OpenReview forum: "GRM-Omni: Generative Omni-modality Reward Modeling via Meta Reward Learning"
_ICLR.cc/2026/Conference — ICLR 2026 Conference Withdrawn Submission_

### Official Review · Reviewer_CXDL · 2025-10-22

**Soundness:** 2
**Presentation:** 1
**Contribution:** 2
**Rating:** 2
**Confidence:** 3

**Summary:**

The paper presents the design of a meta-reward function which combines evaluations from both generative and discriminative reward models (based on foundation models), as well as responses from the policy model to be used for ranking pairs of question-completions (potentially containing multi-modal information, for example images and text). The generative RM infers which critieria justify the selection of a preference in a binary preference dataset and the discriminative RM ranks the importance of the critieria inferred by the generative RM. The meta-reward is used to rank pairs of questions-completions.

**Strengths:**

The reported results show that reward models that are fine-tuned with the rankings provided by the proposed meta-reward achieve great performance in several benchmarks designed to evaluate the quality of reward models. The accuracy on these benchmarks is measured by means of how well can a reward model predict the ground truth preference in a binary preference dataset. Additionally, the proposed meta-reward is a heuristic function that is simple, doesn't require any learning, and can be combined with any off-the-shelf foundation models that act as generative and discriminative reward models.

The paper is significant in that it touches an important area in the field of fine-tuning foundation models with reinforcement learning, which concerns the quality of the reward models used for RLHF. The results show that by improving the design of meta-reward functions over reward models, we can improve these, and in turn the policy resulting from RLHF. The idea of improving reward models beyond learning from static preference datasets is important, and the design of the proposed reward function to do so in this paper is original.

**Weaknesses:**

The paper is generally unclear and hard to follow in several parts.

I find Figure 9 (in the Appendix) to be much more clear to understand the proposed method than not Figure 1 (top) or Figure 5 in the main paper. I find seeing Figure 1 right after the abstract confusing for the reader, as the concepts of TTS, DisRMs or GenRMs have not yet been introduced. Additionally, both left and right panels in Figure 1 have repeated information and it is unclear what the reader should parse from Figure 1 at that point.

Figure 2 is confusing as well, what does it mean that the "Env" outputs a "Scalar" which has an arrow pointing to "DisRM"? What operation is the "Reward Transformation"? What about "Reward Scaling"? These are not described or even defined in the text. I also would not call the proposed reward function a meta-reward, or even less an intrinsic rewards. I find it unclear why the authors introduce intrinsic rewards, explicitly mention curtiosity or motivation since as far as I understand, the proposed approach does not have much to do with the literature of intrinsic rewards in RL.

I think the introduction would be better closed clearly specifying what the proposed method attempts to achieve more clearly: what model is being trained exactly? (e.g. is it the GenRM, the DisRM? the policy?) How is the model being trained? (e.g. SFT? RL? DPO?). As mentioned above, I find a Figure similar to Figure 9 is easier to understand the method than Figure 1 in the introduction.

What is the x-axis in Figure 4? Are they the rankings of the data using the proposed reward function? If so, it would be good to have an example at this point of the paper of differently ranked data points for at least one of this datasets to see how the proposed reward function allows more accurate rankings.

I appreciate the effort in making nice figures for the paper, like Figure 5, but given the clarity of the presented text describing the Figure, it makes it hard for the reader to understand the method. I would recommend either making the FIgure conceptually simplier, or improving the clarity of the description of the figure in the main text, for instance, by enumerating the steps in the flow diagram and explicitly explaining what eahc model and component does and represents at each step.

There are unclear statements in the interpretation of the results (e.g., in Figure 6): "Interestingly, DPO models exhibit a higher lower bound compared to SFT models, suggesting that reinforcement learning can substantially enhance performance on challenging or low-quality samples, thereby improving the overall robustness of the reward model." What evidence is Figure 6 providing on the learning dynamics of foundation models trained with reinforcement learning, if neither SFT or DPO are not reinforcement learning methods?

Finally, some of the main contributions listed in the introduction are not described in detail throughout the paper. For instance, the "criteria-based TTS" technique is not presented rigorously and it remains unclear exactly that contribution does.

I don't find the paper to be ready for publication at the current stage, mainly because the clarity of the presentation, the presence of several wrong/unclear statements, and the lack of detail in the description of the contributions. However, I think that with significant improvements on this points, the idea can be better communicated, the relevance of the results presented can be better appreciated, and the paper can be published.

**Questions:**

What are the x-axis in Figures 4 and 6?

Small error: secnrios in Figure 1 instead of scenarios.

---

### Official Review · Reviewer_JPAR · 2025-10-29

**Soundness:** 3
**Presentation:** 1
**Contribution:** 2
**Rating:** 4
**Confidence:** 3

**Summary:**

This paper introduces GRM-OMNI, which proposes a dual-reward design and meta-reward learning to produce scalable intrinsic rewards, aiming to bridge the gap between sparse environmental rewards and the dynamics of policy learning. GRM-OMNI is an omni-modal reward model that not only achieves strong results on multiple multimodal preference-reward benchmarks but also facilitates more effective policy decisions. In my opinion, this work is interesting, and the blueprint of the reward design is reasonable. However, the paper is somewhat hard to read, and some parts seem problematic or unclear, which I elaborate on in the questions below.

**Strengths:**

- 1. The coupling of DisRM and GenRM is interesting and sound. Similar designs have been introduced in RL reward design, and it is valuable to adapt such a design for LLM training to improve the training process.

- 2. The authors demonstrate strong practical breadth across language, vision understanding/generation, and audio understanding/generation, with promising cross-modal transfer from language-only alignment.

**Weaknesses:**

- 1. Theory: $s_{meta}$ is heuristic. The paper does not provide formal properties such as consistency or guarantees for compounding-bias mitigation. While this may be challenging, please provide some formal properties (e.g., consistency), which are common in reward-design work.

- 2. The correctness of the meta-reward function appears to depend heavily on the reliability of DisRM scores; when DisRM exhibits systematic bias, s_meta may entrench that bias, lacking self-calibration or invariance guarantees.

- 3. I find substantial issues with the writing. Section 3, which should present the overall method, is difficult to follow. Too many concepts are introduced at once without sufficient explanation, especially in Sections 3.2 and 3.3. The authors should use more formal notation to clearly articulate how the proposed method addresses or mitigates the exploration–exploitation trade-off in TTS, and how it achieves “This design allows the reward model to adapt to multimodal inputs during training while preserving its ability to perform CoT reasoning at inference.” If space is limited in the main text, please move the detailed formalization to the appendix.

**Questions:**

- 1. The authors say that DisRM learns a “fast” reward while GenRM learns a “slow” reward, but they also state that DisRMs capture environmental feedback more slowly than GenRMs process scalar rewards. This seems confusing; please clarify it.

- 3. In Section 3.2, how does the proposed method address or mitigate the exploration–exploitation trade-off in TTS?

- 4. In Section 3.3, how does it achieve “This design allows the reward model to adapt to multimodal inputs during training while preserving its ability to perform CoT reasoning at inference”?

---

### Official Review · Reviewer_W8cG · 2025-11-03

**Soundness:** 2
**Presentation:** 1
**Contribution:** 2
**Rating:** 2
**Confidence:** 2

**Summary:**

This paper proposes a mechanism for receiving guidance from both text models and discriminative reward models. It also proposes a somewhat new mechanism of creating diversity in generated outputs.

**Strengths:**

This paper provides an interesting idea; using the disagreement or agreement between a discriminative reward model and a generative reward model to decide the sign of a "meta-reward" function. The results appear to improve slightly over models that leverage existing multi-modal data.

**Weaknesses:**

# Serious
- I think there's a really interesting paper in here probably! But the organization of the paper makes it very challenging to figure out what they're actually doing, how it's implemented, etc. I've tried to provide some concrete suggestions below on how to improve it but it does just need a lot of writing work to be more legible. There's like eight different components being proposed here and they're not particularly organized. It's also not clear which components are important due to missing ablations in the text.

- The paper frequently uses technical terms very loosely in a way that sounds like it has important implications that cannot be understood from the text alone. "However, the step-by-step reasoning of the Dual-RMs is susceptible to compounding bias. That is, the training signals of GenRM cannot be effectively supervised, meaning that false positives may arise even when the predictions appear correct." What is a false positive here? What is compounding bias in this context? As another example "Specifically, the reward model can use $s_\text{meta}$ to sample and prioritize multiple reasoning trajectories... ensuring *consistency* and *causality*". What does causality mean in the context?

# Minor
- The paragraph on line 72 doesn't make too much too much sense; it's discussing properties of the two reward models but they haven't really been introduced yet so the reader can't follow why they have the claimed properties. This issue applies to most of the intro including the subsequent paragraph.
- I'm unsure that Figure 7 represents any improvement over some of the baselines. Hard to tell without any error bars across the rollouts.

# Very minor - does not affect my score but would recommend fixing
- Figure 1 has "secnrios" mispelled.
- I'm not sure it makes much sense to cite OpenAI (2024) Guo (2025) for the reinforcement learning paradigm? I'm not sure under what citation principles that would make much sense. Similarly for citing (Grattafiori et al., 2024; Yang et al., 2025; Team et al., 2025) for LLMs? I don't fully understand the logic behind these citations. They did not introduce these topics.
- Line 87, "dynamic" not "dynamics"
- "RewardBench using Qwen3-32B-AWQ as the policy model and GenRM, the Skywork-Reward model serving as DisRM". This sentence is ambiguous and can be read to imply that the Skywork-Reward model is the GenRM.

**Questions:**

- You have a sentence "Additionally, DisRM cannot extend its capabilities by test-time scaling, as it is constrained by the size of available preference data." This seems not to be true? For reward models based on LLMs, you can apply any test-time scaling technique that you want to improve the quality of the reward model. What is distinct about reward modeling that makes this not true?
- How do you read Figure 4? Why would you want to judge accuracy to go down? What is the x-axis? The highest scoring things get the worst accuracy? I think the reverse is implied but it's hard to extract from the caption. I think a similar issue is occurring in Figure 6?
- You generate multiple candidate criteria $c_i$. How are they used?

---

### Official Review · Reviewer_yED9 · 2025-11-03

**Soundness:** 2
**Presentation:** 2
**Contribution:** 2
**Rating:** 2
**Confidence:** 3

**Summary:**

This paper introduces a meta-reward learning framework to improve reward modeling for preference optimization in large multimodal models. The approach combines a discriminative reward model (DisRM) and a generative reward model (GenRM), connected through a meta-reward function (s_meta) that evaluates the consistency between generated criteria, policy refinements, and reward scores.

The method also proposes criteria-based test-time scaling (TTS), which guides inference using automatically generated evaluation criteria rather than random sampling, enabling more diverse and interpretable reward signals.The resulting system, GRM-OMNI, is trained mainly on language preference data yet generalizes effectively to multimodal benchmarks. Experiments show improvements over existing reward models, and the paper includes analyses of transfer performance and typical failure cases.

**Strengths:**

Novel framework design:
The paper introduces a meta-reward learning framework that bridges discriminative and generative reward models through a well-defined meta-reward function. This design provides a higher-order learning signal to align reward consistency and policy behavior.

Criteria-based reasoning and interpretability:
By introducing criteria-based TTS and language-based evaluation criteria, the approach improves interpretability and enables structured exploration, going beyond black-box scalar rewards.

Multimodal generalization:
The proposed GRM-OMNI shows impressive transfer ability — trained mainly on language preference data but generalizing effectively to multimodal (vision/audio) reward benchmarks.

**Weaknesses:**

The description of how the meta-reward \( s_{\text{meta}} \) is computed is conceptually unclear.  In **Figure 3**, the generative reward model (**GenRM**) is said to produce pairs \( \langle c_i, j_i \rangle \), where \( c_i \) denotes a candidate criterion and \( j_i \) indicates whether the criterion or the judgment is reliable.  Later, however, \( j_i \) is described as reflecting whether GenRM’s judgment is *consistent* with the discriminative reward model (**DisRM**).  This creates a **temporal or causal ambiguity**: at the time GenRM generates \( j_i \), the DisRM scores \( \hat{s}_1^i, \hat{s}_2^i \) are not yet available, so GenRM cannot know that consistency in advance.  It is therefore unclear whether \( j_i \) represents a **predictive judgment** produced by GenRM, or a **post-hoc label** derived after DisRM evaluation.  Clarifying the precise timing, definition, and functional role of \( j_i \) in the computation of \( s_{\text{meta}} \) is essential for understanding how the meta-reward provides meaningful learning signals and whether the framework is **causally well-founded**.

**Questions:**

Please clarify the issues raised in my *Weakness* section — specifically, the conceptual ambiguity around the definition and timing of `j_i` and how it is used in computing \( s_{\text{meta}} \).

---

### Note · Authors · 2026-01-07

I have read and agree with the venue's withdrawal policy on behalf of myself and my co-authors.